# Prevalence of Respiratory Health Symptoms among Landfill Waste Recyclers in the City of Johannesburg, South Africa

**DOI:** 10.3390/ijerph16214277

**Published:** 2019-11-04

**Authors:** Nonhlanhla Tlotleng, Tahira Kootbodien, Kerry Wilson, Felix Made, Angela Mathee, Vusi Ntlebi, Spo Kgalamono, Moses Mokone, Karen Du Preez, Nisha Naicker

**Affiliations:** 1National Institute for Occupational Health, National Health Laboratory Services, Johannesburg 2000, South Africa; TahiraK@nioh.ac.za (T.K.); KerryW@nioh.ac.za (K.W.); Felixm@nioh.ac.za (F.M.); vusiNT@nioh.ac.za (V.N.); spok@nioh.ac.za (S.K.); Mosesm@nioh.ac.za (M.M.); KarenD@nioh.ac.za (K.D.P.); Nishan@nioh.ac.za (N.N.); 2School of Public Health, Faculty of Health Sciences, University of the Witwatersrand, Johannesburg 2000, South Africa; angie.mathee@mrc.ac.za; 3Department of Environmental Health, Faculty of Health Sciences, University of Johannesburg, Johannesburg 2000, South Africa; 4Environment and Health Research Unit, South African Medical Research Council, Johannesburg 2000, South Africa

**Keywords:** Cross-sectional study, informal workers, landfill sites, respiratory symptoms, waste recyclers

## Abstract

In developing countries, waste sorting and recycling have become a source of income for poorer communities. However, it can potentially pose significant health risks. This study aimed to determine the prevalence of acute respiratory symptoms and associated risk factors for respiratory health outcomes among waste recyclers. A cross-sectional study was conducted among 361 waste recyclers at two randomly selected landfill sites in Johannesburg. Convenience sampling was used to sample the waste recyclers. The prevalence of respiratory symptoms in the population was 58.5%. A persistent cough was the most common symptom reported (46.8%), followed by breathlessness (19.6%) and rapid breathing (15.8%). Approximately 66.4% of waste recyclers reported exposure to chemicals and 96.6% reported exposure to airborne dust. A multivariable logistic regression analysis showed that exposure to waste containing chemical residues (OR 1.80, 95% CI 1.01–3.22 *p* = 0.044) increased the odds of respiratory symptoms. There was a significant difference in respiratory symptoms in landfill sites 1 and 2 (OR 2.77, 95% CI 1.03–7.42 *p* = 0.042). Occupational health and safety awareness is important to minimize hazards faced by informal workers. In addition, providing waste recyclers with the correct protective clothing, such as respiratory masks, and training on basic hygiene practices, could reduce the risks associated with waste sorting.

## 1. Introduction

With increasing volumes of solid waste, and a growing global market for recycled materials, it is estimated that about 15 million people in low and middle income countries work as waste recyclers [1]. Waste recyclers earn a living by collecting and selling recyclable items of municipal solid waste. Most waste recyclers sort the waste without the use of personal protective equipment (PPE), thus exposing themselves to a range of environmental health hazards [2]. While this type of work provides a source of income, waste recyclers face some health hazards and safety risks, including injuries, gastrointestinal problems, infections and respiratory conditions [3]. In addition to the occupational health risks, poor living conditions, hygiene practices and lifestyle habits increase their risk of acquiring diseases [1].

Previous studies have reported associations between working at landfill sites and increased health risks, such as musculoskeletal disorders of the lower back, shoulders and neck [4]; upper and lower respiratory symptoms, with a high prevalence of coughs [5]; and mental health disorders [6]. A review by Binion and Gutbertlet (2012) reported an increased incidence of respiratory diseases among waste recyclers exposed to dust, fumes from chemicals and inhalation of sulphurous gases generated by anaerobic microbial decomposition of organic waste [7]. In addition, a study in Kaohsiung County, Taiwan, found waste recyclers exposed to chemical residues from household cleaning detergents, paint and pesticide containers reported acute respiratory symptoms, such as coughing and shortness of breath, headaches, sinusitis, and nausea [8].

The health of landfill workers is affected by socio-economic factors, such as education status, living close to landfill sites, informal housing and the use of solid biomass fuels for cooking, such as wood and coal, which have been shown to increase inhalable particulate materials, thus increasing the risk of chronic respiratory symptoms [4,9]. There are few studies investigating health outcomes associated with waste picking in the informal economy. The scarcity of research on the health of informal workers in the South African context has necessitated that more research is done on informal waste recyclers. The objective of this study was to evaluate the prevalence of acute respiratory symptoms among waste recyclers at two major landfill sites in Johannesburg. The study will further identify risk factors associated with respiratory symptoms.

## 2. Materials and Methods

### 2.1. Study Setting and Population

This study was a cross-sectional study conducted at two landfill sites, situated in the west and southwest of Johannesburg. There are four Pikitup landfill sites in Johannesburg, Gauteng, South Africa, where municipal waste is disposed. Two of the largest landfill sites were selected for the study. Landfill site 1 hosted approximately 3000 waste recyclers and landfill site 2 had approximately 600 waste recyclers. The study population consisted of male and female informal waste recyclers over 18 years of age. Waste recyclers who were available during the study days were invited to participate in the study. A survey sample size calculation provided a sample size of 365, with a confidence level of 95% and a significance level of 0.05. Using proportional sampling strategy, 82% (*n* = 299) of waste recyclers were recruited from landfill site 1 and 17% (*n* = 62) were recruited from site 2. Approval was sought from the landfill management, and ethical clearance was obtained from the Human Research Ethics Committee of the University of the Witwatersrand (clearance number M171120) to conduct the study.

The study was divided into two phases. In phase I, a qualitative health risk assessment of the two landfill sites was conducted by a trained occupational hygienist, in order to assess safety practices and to identify hazards associated with the work performed. The risk assessment process based on the “Five steps to risk assessment,” developed by the UK Health and Safety Executive (HSE) [10], was followed. In phase II, waste recyclers were interviewed by trained nurses using a structured questionnaire.

### 2.2. Study Variables

#### 2.2.1. Socio-Economic Variables

Data pertaining to the participants’ personal information were collected. These included age, sex, education, number of years spent working as a waste recycler, and source of fuel used for cooking, which determined whether they used electricity, paraffin, gas, wood/coal or any other source. Behavioral characteristics collected included a history of cigarette smoking (whether participants currently, or had ever, smoked) and number of years smoking. The waste recyclers were asked whether they thought working at a landfill site affects their chest. The question was phrased as, “Do you think working at a landfill site affects your chest: Yes or No”. Additional information was also collected on their medical history, including whether they had previously been diagnosed with TB or asthma, and whether they had a history of any muscle sprains and strains.

#### 2.2.2. Explanatory and Dependent Variables

The variables that were hypothesized to be associated with respiratory symptoms were exposure to airborne dust on the landfill sites, and handling chemical waste, such as household detergents, paint and pesticide containers. Participants were asked whether they used any form of personal protective wear. To collect data on these variables, participants were asked if airborne dust was (1) a “major problem or moderate problem” or (0) “no problem”. Use of PPE (mask, boots or gloves) was classified as (0) “always”, (1) “sometimes or never”. Exposure to chemical waste (cleaning detergents, paint) was classified as a binary variable (“Yes” if participants were exposed and “No” if unexposed). The dependent variable was self-reported respiratory symptoms, defined as “Yes” if a participant reported at least one of the following symptoms: a persistent cough, coughing with blood, wheezing or whistling in the chest, breathlessness and rapid breathing. It was defined as “No” if none of the symptoms were reported.

#### 2.2.3. Confounders

Age, number of years spent working at the landfill site, smoking status (current smoker/history of smoking), a diagnosis of TB or asthma, and source of fuel for cooking were considered as possible confounding variables.

### 2.3. Data Analysis

The data were collected in real time using Research Electronic Data Capture (REDCap^®^). Data cleaning and analysis were conducted using Stata Statistical Software: Release 15 SE (StataCorp. 2017. College Station, TX: StataCorp LLC). Data cleaning involved removing observations with missing data and observations where data were incorrectly completed. A dataset with 361 observations was then used for analysis. Median and interquartile ranges (IQR) were used, along with the non-parametric Mann Whitney rank-sum test, to describe differences. Pearson chi-squared and Fishers exact tests were used to assess the association between categorical variables. A multivariable logistic regression was used to calculate the crude and the adjusted odds ratio at 95% confidence intervals (CI). Model building was used to choose variables that were included in the final model. Variables that were statistically significant and those that are considered important based on the existing literature were included in the final model. For the final adjusted model, a stepwise logistic regression was used to include variables at the 5% level. The final model was adjusted for all confounding variables and was stratified by landfill sites. Statistical interaction between independent variables in the final model was investigated.

## 3. Results

A total of 361 conveniently sampled participants completed the survey. Of these, 292 (83%) were from landfill site 1, and 62 (17%) were from landfill site 2 (Table 1). The study population was comprised of 259 (73.7%) males, with the largest proportion aged 29–39 years (median age = 31 years, IQR, 27–39). A total of 274 (77.6%) of the waste recyclers had achieved a secondary school education level, with 5 (1.4%) having acquired tertiary education. A total of 341 (95.55%) of the waste recyclers reported always wearing PPE, with 4.45% reporting wearing PPE sometimes or never. Approximately 220 (75.3%) waste recyclers in landfill 1 reported a history of smoking, compared to 26 (41.9%) in landfill site 2. Electricity was reported as the major source of fuel for cooking 239 (67.5%), while a number of participants used paraffin 65 (19.1%) and wood 39 (11.02%) as sources of fuel. The median number of years the workers had worked in the landfill areas was 5 (IQR, 3–10). The chi square test showed a significant difference between the two landfill sites in education status, self-reported smoking history, and the number of years spent working in a landfill sites (*p* < 0.001). Data on exposure variables, chemical and airborne dust showed that 159 (47.9%) of the landfill workers reported inhaling fumes. Approximately 194 (66.7%) of the workers in landfill site 1, and 21 (35%) in landfill site 2 were exposed to chemicals. Overall, 340 (96.6%) reported airborne dust as a major problem in the landfill sites.

Table 2 shows occupational exposures faced by waste recyclers in landfill sites. Airborne dust from soil and waste material, as well as chemicals from household waste, were reported as the main sources of chemical exposure for waste recyclers.

The prevalence of respiratory symptoms in the study population is described in Table 3. The overall prevalence of respiratory symptoms in the study was 58.5%. Nearly half of the landfill workers reported having a persistent cough. Landfill 1 showed a significantly higher proportion of self-reported persistent coughs, at 163 (46.8%; *p* < 0.05) of the participants, while there was no significant difference between landfills for the rest of the symptoms. Breathlessness was the second most prevalent symptom with 56 (19.4%) participants in landfill 1, and 13 (20.9%) participants in landfill site 2 reporting breathlessness as a respiratory symptom. About 55 (15.8%) and 48 (13.7%) waste recyclers reported symptoms of rapid breathing and wheezing. Approximately 10 (2.8%) of the study participants reported coughing blood.

The crude and the adjusted odds ratios are shown in Table 4. Univariable analysis of respiratory symptoms and study variables showed that waste recyclers at landfill site 1 had 1.78 times greater odds of reporting respiratory symptoms, and the association was statistically significant. Reporting a previous or current history of smoking (unadjusted OR 1.65, 95% CI 1.04–2.60) showed a significant association with respiratory symptoms. Chemical exposure was also statistically associated with respiratory symptoms in the univariable analysis.

The multivariable analysis, adjusting for known correlates of respiratory symptoms, showed that waste recyclers at landfill site 1 had increased odds of respiratory symptoms, AOR 2.77 (95% CI 1.03; 7.42). Chemical exposure was shown to have a statistically significant association with respiratory symptoms, while those who reported being current smokers, or having a history of smoking, showed a marginally significant association. The adjusted odds of having respiratory symptoms for waste recyclers exposed to chemicals in landfill site was 1.80 (1.01–3.22), and those who currently or previously smoked had 3.52 times greater odds of reporting respiratory symptoms (95% CI 0.92–15.54). No significant difference was found with the other main predictors (use of PPE, airborne dust, source of fuel for cooking, years working at landfill site) of the study outcome. There was no statistical interaction between the independent variables in the final model.

## 4. Discussion

Adverse health effects in waste recyclers have been studied in other developing countries [4,11], but there is limited research on this sector in South Africa. The health risks and injuries that are incurred by waste recyclers are a public health concern, which needs attention from all relevant stakeholders. In this study, the prevalence and risk factors associated with respiratory symptoms among landfill waste recyclers in Johannesburg municipality were investigated. In the adjusted multivariable analysis, socio-demographic characteristics (age, sex, educational level) and other predictors, such as number of years spent working as a waste recycler, source of fuel for cooking and use of PPE, did not show a significant association with reported respiratory symptoms.

Overall, the results of this study showed a high prevalence of coughing as a respiratory symptom among waste recyclers, compared to other self-reported symptoms, such as wheezing, breathlessness and rapid breathing. Similar results were found in other studies. For example, a study assessing the respiratory health effects of landfill workers, using spirometric lung function, found a high prevalence of respiratory symptoms, such as coughing, wheezing and chest discomfort in waste pickers, compared to workers who did not work on landfill sites [5]. Similarly, Chokandre et al (2017) found a high prevalence of breathing difficulties and a chronic cough among waste pickers of Mumbai, India [12]. Oyelola et al (2011) also found coughing to be the major respiratory health problem for waste pickers working at landfill sites in Lagos metropolis [13].

Waste recyclers exposed to chemicals had increased odds of developing respiratory symptoms, as illustrated in the results of the final model. Examples of chemical waste may include containers of domestic cleaning detergents, paints, batteries, electronic equipment and solvents. Methane, and other gases emitted from decomposition of waste products or chemical bottles, may lead to respiratory symptoms [14]. It is known that occupational exposure to vapor, gases, dusts, and fumes affects large airway function and increases the risk of spirometry-defined COPD [15,16].

The analysis showed that landfill workers exposed to airborne dust reported twice the odds of respiratory symptoms, compared to those not exposed. Although our findings are not statistically significant, related studies found that the majority of waste pickers complained of bad smell and dust from the landfill, which they indicated affects their health [17,18]. Vehicle fumes from trucks off-loading the waste and burning waste has been shown to be the major cause of airborne dust in landfill sites [19]. In addition, the poor working conditions of waste recyclers, who are daily exposed to dust, infectious bacteria, gases and bio-aerosols, have also been shown to cause respiratory health problems [10]. Further studies are necessary to measure respirable quartz in airborne dust, which may cause silicosis, chronic obstructive pulmonary disease or lung cancer, upon prolonged or repeated inhalation exposure [20].

Smoking increases the prevalence of respiratory symptoms [5,21]. The prevalence of smoking is higher in low socioeconomic groups, especially among disadvantaged individuals, such as informal workers [22,23]. In our analysis, the odds of reporting respiratory symptoms for those with a history of smoking was almost four times greater compared to those who did not report a history of smoking. In addition, smoking was found to be marginally associated with an increased prevalence of respiratory symptoms. Smoking is a well-known confounder of respiratory symptoms [5,21]. The negative effects of occupational exposure have been shown to be pronounced in those with a history of smoking [21].

We reported differences between landfill sites in the prevalence of respiratory symptoms. The final analysis showed a significant difference between landfill sites and prevalence of respiratory symptoms. There were differences in work practices and the condition of work between the two landfill sites, observed during the risk assessment walkthrough. Even though the majority of waste recyclers in the two landfill sites reported using PPE during work, it was observed that the work was carried out with no proper or occupationally recommended PPE, such as N95 masks. Some of the waste recyclers did not use any form of protective wear. Inappropriate use of PPE may enhance the susceptibility of these waste recyclers to health problems [4,20]. Lack of awareness of the importance of proper protective clothing might be attributed to the waste workers’ lack of education on occupational health, and the dearth of knowledge on hazards associated with waste picking [17].

While this study offers a preliminary overview of the respiratory health effects of waste pickers in Johannesburg, any generalization of the results must be done with caution, as only landfills in Johannesburg municipality were chosen. In addition, study participants were selected on convenience, rather than through random selection. Recall bias may also have affected the estimated prevalence of the outcome, thus, authors recommend a complementary study with spirometry and exposure assessment, to strengthen the findings in this study. Although the study’s sample size could be improved, it was very close to the 365 previously calculated as sufficient, and can be reported as a strength of the study.

## 5. Conclusions

The health of waste recyclers is a public health concern, due to the number of risk factors associated with this activity. The results of this study show that exposure to chemical waste increases the risk of reporting respiratory symptoms, such as coughing and breathlessness. Poor work practices and lack of proper PPE for waste recyclers is a factor contributing to the adverse health effects experienced by this class of worker. As indicated, waste recycling is becoming a source of income for unemployed communities, thus, occupational health and safety measures should be put into place to minimize hazards faced by landfill waste recyclers. Provision of appropriate PPE half-mask respirators, fitted with ABEK-P3 combination filters, and training on health hazards associated with waste picking, including the spread of harmful bacteria and respiratory effects, could reduce risk factors associated with waste picking. In addition, health and safety awareness, including training on basic hygiene, should be provided to reduce the risks associated with waste sorting.

## Figures and Tables

**Table 1 ijerph-16-04277-t001:** Description of the socio-demographic characteristics of the study population and risk factors associated with respiratory symptoms.

Characteristics	Total (n, %)	Landfill Site 1 (*n* = 292)	Landfill Site 2 (*n* = 62)	*p* Value
Sex				
Male	259 (73.16)	229 (78.42)	30 (48.39)	<0.001
Female	95 (26.84)	63 (21.58)	32 (51.61)	
Age				
18–28	123 (34.75)	118 (40.41)	5 (8.06)	<0.001
29–39	152 (42.94)	129 (44.18)	23 (37.10)	
40–50	47 (13.28)	29 (9.93)	18 (29.03)	
51+	32 (9.04)	16 (5.48)	16 (25.81)	
Education				
None	15 (4.25)	11 (3.78)	4 (6.45)	<0.001
Primary	59 (16. 71)	35 (12.03)	24 (38.71)	
Secondary	274 (77.62)	241 (82.82)	33 (53.23)	
Tertiary	5 (1.42)	4 (1.37)	1 (1.42)	
Current/Ever Smoked				
Yes	246 (69.49)	220 (75.34)	26 (41.94)	<0.001
No	108 (30.51)	72 (24.66)	36 (58.06)	
Years smoked	10 (6–15)	median 10: IQR 6–14	median 17: IQR 10–21	<0.001
Occupation				
Years Working in Landfill Site	5 (3–10)	median 4: IQR 2–7	median 13: IQR 9–17	<0.001
Perception that Landfill Affects Chest				
Yes	158 (45.40)	134 (46.90)	24 (39.34)	0.295
No	190 (54.60)	153 (53.31)	37 (60.66)	
Medical History				
% Tuberculosis	11 (3.14)	6 (2.08)	5 (8.06)	0.014
% Asthma	12 (3.40)	12 (4.12)	0 (0.00)	0.104
% Sprains and Muscle Strains	98 (29.25)	80 (28.88)	18 (31.03)	0.743
Source of Fuel (Cooking)				
Electricity	239 (67.51)	185 (63.36)	54 (87.10)	0.010
Paraffin	65 (18.36)	59 (20.21)	6 (9.68)	
Gas	5 (1.41)	5 (1.71)	0 (0.00)	
Wood/coal	43 (12.15)	41 (14.04)	2 (3.23)	
Other	2 (0.56)	2 (0.68)	0 (0.00)	
Use Personal Protective Equipment (PPE) (%)				
Always	341 (96.33)	279 (95.55)	62 (100.00)	0.087
Sometimes/Never	13 (3.67)	13 (4.45)	0 (0.00)	
Exposures				
Chemicals (%) (*n* = 351)				0.803
Yes	233 (66.38)	194 (66.67)	21 (35.00)	
No	118 (33.62)	97 (33.33)	39 (65.00)	
Airborne Dust (%) (*n* = 352)				0.007
No problem	12 (3.41)	6 (2.07)	6 (9.68)	
Moderate/major problem	340 (96.59)	284 (97.9)	56 (90.33)	

**Table 2 ijerph-16-04277-t002:** Description of occupational exposures and hazards common on landfill sites, assessed using a health risk assessment process.

Exposures/Hazards	Observation	Landfill Site 1	Landfill Site 2
Airborne Dust	Waste reclaiming workers may be exposed to dust liberated from the soil, and waste material, by the dump truck and compactor. The dust may contain organic matter, which may cause skin or respiratory irritant or allergic reactions, or contain pathogens.	The majority of the waste reclaimers did not wear dust masks to protect themselves from dust inhalation while extracting recyclable materials.	A water truck is used for wetting the soil on the roadways where dump trucks are operating.Some reclaiming workers were using garments or scarves to cover their mouth and nose as protection against dust inhalation.
Hazardous Chemical Substances (HCS): Organic Dust, Pesticides or Organic Solvents	Waste reclaiming workers are exposed to various classes of chemical substances in general household waste.	No adequate control measure to the waste recyclers to protect against inhalation of toxic fumes. The majority of the waste reclaimers did not wear proper masks during work in the landfill site.	Waste recyclers were seen using garments or scarves to cover their mouth and nose as protection against dust inhalation.

**Table 3 ijerph-16-04277-t003:** Prevalence of self-reported respiratory symptoms stratified by landfill site.

Respiratory Symptoms	N	Total Reported Yes *n* (%)	Landfill Site 1	Landfill Site 2	*p*-Value
Persistent Cough	348	163 (46.8)	145 (50.7)	18 (29.0)	0.002
Coughing blood	351	10 (2.8)	10 (3.4)	0	0.137
Wheezing	350	48 (13.7)	41 (14.0)	7 (11.2)	0.541
Breathlessness	351	69 (19.6)	56 (19.4)	13 (20.9)	0.775
Rapid Breathing	349	55 (15.8)	46 (15.6)	9 (14.5)	0.813

**Table 4 ijerph-16-04277-t004:** Crude and adjusted odds ratios from univariable and multivariable analysis of study variables and respiratory symptoms.

Characteristics	Respiratory	No Respiratory	Unadjusted OR (95% CI)	*p*	Adjusted OR (95%CI)	*p*
Landfill Sites						
Landfill Site 1	178 (60.96)	114 (39.04)	1.78 (1.02; 3.08)	0.041	2.77 (1.03; 7.42)	0.042
Landfill Site 2	29 (46.77)	33 (53.23)	ref			
Sex						
Male	164 (77.0)	101 (68.24)	0.64 (0.40; 1.02)	0.065	0.88 (0.34; 2.25)	0.793
Female	49 (23.0)	47 (31.76)	ref			
Age						
18–28	79 (37.1)	47 (31.76)	1.48 (0.67; 3.24)	0.324		
29–39	90 (42.25)	65 (43.9)	1.22 (0.57;2.62)	0.607		
40–50	22 (10.33)	16 (10.81)	1.13 (0.46; 2.79)	0.783		
51+	22 (10.33)	20 (13.51)	ref			
Education						
None	10 (4.72)	5 (3.38)	0.75 (0.93; 6.04)	0.787		
Primary	36 (16.98)	23 (15.54)	0.78 (0.24;2.58)	0.687		
Secondary	163 (76.89)	118 (79.73)	0.69 (0.23; 2.07)	0.509		
Tertiary	3 (1.42)	2 (1.35)	ref			
Current/Ever Smoked						
Yes	158 (62.70)	94 (37.30)	1.65 (1.04; 2.60)	0.031	3.52 (0.92; 15.54)	0.066
No	55 (50.50)	54 (49.50)	ref			
Years Smoked (Median, IQR)	10 (6–14)	17 (10–21)	0.97 (0.94; 1.05)	0.097	0.96 (0.93; 1.01)	0.085
Occupation (Median, IQR)						
Years Working in the Landfill Sites, Median (IQR)	4 (2–7)	13 (9–17)	0.99 (0.95; 1.03)	0.844	1.03 (0.965; 1.109)	0.333
Perception that Landfill Affects Chest						
Yes	112 (53.08)	51 (35.42)	2.06 (1.33; 3.19)	0.001	1.54 (0.89; 2.66)	0.123
No	99 (46.92)	93 (64.58)	ref			
Medical History						
% Tuberculosis	7 (63.60)	4 (36.40)	1.21 (0.35; 4.23)	0.756		
% Sprains and Muscle Strains	69 (34.85)	30 (21.28)	1.97 (1.20; 3.26)	0.007	1.53 (0.89; 2.67)	0.877
Source of Fuel						
Electricity	133 (55.20)	108 (44.80)	ref			
Paraffin	45 (65.20)	24 (34.80)	1.52 (0.87; 2.66)	0.139	1.77 (0.83; 3.74)	0.138
Gas	2 (33.30)	4 (66.70)	0.41 (0.72; 2.26)	0.303	0.27 (0.27; 2.86)	0.281
Wood/Coal	32 (15.0)	11 (7.40)	2.36 (1.13; 4.90)	0.021	1.59 (0.70; 3.63)	0.264
Other	1 (50)	1 (50)	0.81 (0.05; 13.13)	0.883	0.72 (0.43; 12.25)	0.822
PPE (%)						
Always	174 (56.10)	136 (43.90)	ref	0.604		
Sometimes/Never	23 (63.90)	13 (36.10)	0.69 (0.27; 1.77)	0.445	0.74 (0.45; 1.22)	0.239
Exposures Chemicals (%)						
Yes	154 (72.99)	82 (55.78)	2.14 (1.37; 3.34)	0.001	1.80 (1.01; 3.22)	0.044
No	57 (27)	65 (44.22)	ref			
Airborne Dust (%)						
No problem	5 (41.70)	7 (58.30)	ref			
Moderate/Major Problem	201 (57.60)	146 (42.10)	2.07 (0.64; 6.65)	0.222	2.50 (0.30; 20.65)	0.394

*Hosmer-Lemeshow chi2 (8)* = 3.75, *p* = 0.8790. Correlation coefficient between age and years working in landfill site was 0.559. Final model was adjusted for age, years working at the landfill site, smoking status (current smoker/ever smoked), history of TB and asthma, and source of fuel for cooking.

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
