# Peer review of "Prevalence of Respiratory Health Symptoms among Landfill Waste Recyclers in the City of Johannesburg, South Africa"

_ijerph, 2019, doi:10.3390/ijerph16214277_

Round 1

Reviewer 1 Report

Specific comments:

Abstract:

More detailed information on methods in the abstract would be beneficial.  How were these 361 participants selected? Even how these two sites have been selected? This should be clear in the method section of the abstract.  How the authors defined respiratory symptoms? Are those have reported by participants? Is that self-reported or doctor-diagnosed?  What variables are considered as a confounder in the statistical analysis? 

Introduction:

I missed a rationale for doing this research? What’s the gap in this field of research? How many previous studies have reported acute respiratory symptoms? What’s the variations between studies in terms of study design, geographic locations, sample size etc. This should be highlighted in the intro. I’m not convinced with author’s comments in the last line of intro. (line 62, page 2 of 12).

Methods: 

Reference required for HSE five steps risk assessment tool (page 2 of 12, line 74).  Page 2 of 12, lines 77-78 is not clear.  A more detailed operational definition would be required. The second paragraph (page 2 of 12, lines 81-91) is not very clear and helpful to follow. How many steps were followed in the data collection process? Two steps? Page 2 of 12, line 71 stated “qualitative health risk assessment of the two landfill sites was conducted by a trained occupational hygienist” while line 81 stated, “ Waste recyclers were interviewed by trained nurses using a structured questionnaire”. I found this is confusing and conflicting. Can you please make it clear and justify these two steps in data collection?  As two centres were included, I think it’s better to adjust additionally for centre variable.

Results: 

Why there are large variations in terms of participation in the two sites (n=292 vs n=62)?

Discussion: 

Again, what are the limitations in research in South Africa? This should be highlighted in the first paragraph.  Page 4 of 12, lines 193-200, paragraph 3, is not well justified. Need to clarify why the results are different compared to previous studies.  Not sure, how participants are aware of airborne dust? Airborne dust is very difficult to measure and required technical assistance to quantify.  There are many limitations in this study, including self-reported outcome, self-reported exposure, a random selection of participants and chance of recall bias. All this should acknowledge in the limitation section in the discussion. Currently, this is missing.  Conclusion should be more focused and robust.

Author Response

Reviewer 1

Comment

Addressed (Yes/No)

comment is addressed/Justification

Page and line number

Abstract

More detailed information on the methods in the abstract.

1.Indicate how the 361 participants were selected.

2. Indicated how the study sited were selected.

3. Indicate how respiratory symptoms were defined.

4. Indicate variables considered as confounders.

Yes

Yes

Yes

Yes

The minimum required words in the abstract are 200 thus not all comments suggested by the review could be included in this section. These are however discussed in detail in the methods section.

Convenience sampling was used to sample the waste recyclers

“A cross sectional study was conducted among 361 waste recyclers in two randomly selected landfill sites in Johannesburg”.

Indicated in the methods section

“The dependent variable was self-reported respiratory symptoms defined as “Yes” if a participant reported at least one of the following symptoms: a persistent cough, coughing with blood, wheezing or whistling in the chest, breathlessness and rapid breathing and “No” if none of the symptoms were reported”

Indicated in the methods section

“The final model was adjusted for age, years of working at the landfill site, smoking status (current smoker/ever smoked), being diagnosed with TB and asthma, source of fuel for cooking (electricity, paraffin, gas, wood, coal and other)”

Page 1 of 12, line 19

Page 1 of 12, line 19

Page 3 of 12, line 105-108

Page 3 of 12, line 121-124.

Introduction

1.Indicate rationale for the research

2.Whats is the gap in the field of research.

3.Previous studies that have reported acute respiratory symptoms.

Yes

Yes

Yes

Indicated in the text:

“There are few studies investigating health outcomes associated with waste picking in the informal economy”.

Indicated in the text “Thus, the scarcity of research on the health of informal workers in the South African context has necessitated that more research is done in informal waste recyclers”

These studies are referenced in the introduction.

Referenced in the introduction: “Previous studies have reported associations between working at landfill sites and increased risks to health such as musculoskeletal disorders of the lower back, shoulders and necks [4], upper and lower respiratory symptoms with high prevalence of cough [5] and mental health disorders [6]. A review by Binion and Gutbertlet (2012) reported an increased incidence of respiratory diseases among waste recyclers”

Page 2 of 12, line 56-57

Page 2 of 12, line 57-59

Page 2 of 12, line 43-47

Methods

1.Reference is required for the HSE five step risk assessment.

2.Page 2 of 12, line 81-91 not clear to follow

3. Page 2 of 12, line 71.

4. Adjust for centre variable

Yes

Yes

Yes

The risk assessment process based on the “Five steps to risk assessment” developed by the UK Health and Safety Executive (HSE)

Addressed: “The study was divided into two phases. Phase I, a qualitative health risk assessment of the two landfill sites was conducted by a trained occupational hygienist…”

“In phase II, waste recyclers were interviewed by trained nurses using a structured questionnaire…..”

Indicated in the methods section that the analysis was stratified by landfill sites.

Page 2 of 12, line 81-83.

Page 2 of 12, line 79-84.

Page 3 of 12, line 123-124.

Results

1.Why there are large variations in terms of participation in the two sites (n=292 vs n= 62)

Yes

The sample size for each landfill site was selected based on the number of waste recyclers in the landfill site by proportional sampling strategy.

Methods section: “Landfill site 1 hosted approximately 3000 waste recyclers and landfill site 2 had approximately 600 waste recyclers.

Page 2 of 12, line 69-70.

Discussion

1.What are the limitation in research in South Africa.

2.Page 4 of 12, line 193-200, para 3 not well justified.

3.Clarify why results are different compared to previous studies.

4.State limitations clearly (including self-reported outcome, self-reported exposure, a random selection of participants and chance of recall bias)

Yes

Yes

Yes

Yes

Added in introduction paragraph of discussion “The health risks and injuries that are incurred by waste recyclers are of public health concern that needs attention from all relevant stakeholders”.

Removed. “Despite the use of wood and coal as a source of biomass fuel not being significantly associated with respiratory symptoms in the final model, other studies have shown that waste recyclers who used wood and other recyclable items as fuel for cooking were significantly more likely to develop respiratory illness compared to those who used liquefied petroleum gases (LPG)[14].

The results of the study may be different due to limitations of the study as indicated in the discussion.

Indicated in the last paragraph of the discussion “While this study may give a preliminary overview of respiratory health effects of waste pickers in Johannesburg, generalization of the study results must be done with caution as only landfills in Johannesburg municipality were chosen. In addition, study participants were selected on convenience rather than random selection. Recall bias may also have affected the estimated prevalence of the outcome; thus authors recommend a complementary study with spirometric and exposure assessment to strengthen the findings in this study. Even though the study's sample size can be improved, it was very close to the 365 previously calculated as sufficient, and can be reported as a strength of the study.”

Page 4 of 12, line 190-192.

Page 5 of 12, line 240-247.

Page 5 of 12, line 240-247.

Conclusion

Should be more focused and robust

Yes

Added to the conclusion “The results show that exposure to chemical waste increases the risk of reporting respiratory symptoms such as coughing and breathlessness. Poor work practices and lack of use of proper PPE for waste recyclers is a factor contributing to the adverse health effects experienced by this class of workers”

Page 5 of 12, line 249-259.

Reviewer 2 Report

The study is basic, mostly superficial and, in my opinion, it is rather poorly presented.

The conclusions refer to potential implications of the study, but they are quite disjointed from the results themselves, where the effect of using masks or respirators was not considered. The conclusions should be directly supported by the results, not a speculation of what the results may (or may not) suggest.

The methods could have included some air quality samples, and some respiratory tests in a subset of participants, to demonstrate some actual effects of the exposure. This evidence would very much strengthen the novelty of the work, its scientific rigour and appeal.

Statistical analysis, line 101: provide a justification for the "5% level". Why was this level chosen?

Results: why are the participants "conveniently sampled"?

Reduce the number of tables and data in the core part of the manuscript to maintain focus on the main results.

Expand table legends to be more informative.

Reduce the length and speculation level in the discussion.

Most of the literature cited was published in the last few years. Please include more original studies, and not merely more recent ones (some of which are confirmatory rather than novel).

I hope that the above mentioned points may help improve the manuscript.

Author Response

Reviewer 2

Comment

Addressed (Yes/No)

comment is addressed/Justification

Page and line number

Conclusion should be supported by the results

Yes

The results show that exposure to chemical waste increases the risk of reporting respiratory symptoms such as coughing and breathlessness. Poor work practices and lack of use of proper PPE for waste recyclers is a factor contributing to the adverse health effects experienced by this class of workers”

Page 5 of 12, line 249-259.

Results

Statistical analysis, line 101: provide a justification for the “5% level”

Yes

The response is not added to the paper, nonetheless: The significance level indicates the probability of rejecting the null hypothesis when it is true. In statistics, we use 0.05 or 5 % to reject the null hypothesis.

Results:

Why are the participants “conveniently sampled”

Yes

The response is not added to the paper. We used convenience samples because a random sample could be done due to the lack of access of the waste pickers , hence any waste picker who was available on the day of the study they were invited to participate.

Results

Reduce the number of tables and data to maintain focus on the main results.

No

The comment was considered, nonetheless, all the tables contain relevant information for the objective of the paper that cannot be omitted.

Results

Expand table legend to be more informative

Yes

Comment addressed

Discussion

Reduce the length and speculation level in the discussion

Yes

Paragraph 3 and 4 were removed from the discussion.

Page 4 of 12.

Reviewer 3 Report

This is an interesting study assessing the respiratory symptoms among landfill workers in Johannesburg, where authors presented the associations of symptoms with risk factors collected through questionnaire, and qualitative data about the exposure pathways. Although, significant statistical and data presentation/discussion errors strongly compromise the quality of the paper, and must be addressed before accepted for publication. Some comments in each section are presented below:  

General questions: A new economic status classification was proposed to became less pejorative, which is low-, middle-, and high-income countries. I would suggest authors to replace "developing countries" by lower-income or low- and middle-income countries.

Self-reported respiratory symptoms were dichotomized in yes/no, but would be very interesting if authors have tested for other associations, such as what exposure factors are associated with each symptom and 2 or more symptoms.

Authors alternate the data presentation data throughout the text; e.g. n(%) and %(n). I suggest standardizing it.

The article presents some grammar and typo errors that must be double checked before the publication.

Abstract: Authors affirmed that "66.4 % of waste recyclers reported exposure to chemicals" but we are all exposed. Will be nice if they can specify which chemicals or how they are exposed to.

Methods/Data analysis: In data analysis section, authors presented many important variables used in the final models that were not mentioned before, such as age, years working at landfill site, smoking status, living close to the landfill site, TB and asthma diagnose, and source of fuel for cooking. These variables must be described in methods. Moreover, is the use of PPE refering only to respiratory PPE, any PPE or at least one? 

Results: Table 1 present the p-values as 0.000 but it must be replaced by <0.001. Also, authors should standardize data presentation, and if classification is presented in the first column it does not need to be repeated (e.g. median, IQR). The variable presented in Table 1 "Landfill affects chest" is not described and explained in the methods. Authors used Chi-square to test for associations between groups but I would recommend using Fisher exact test for variables with less than 5 observations (e.g. source of fuel, education). Also, consider grouping variables to increase the statistical power of associations (e.g. source of fuel). Please, consider adding some footnotes do table 1 to explain the abbreviations and other relevant information. In Table 1, some associations seems incorrect (p-values), for instance, significant differences between groups were observed for airborne dust (98% vs. 90%), but not for chemicals (57% vs. 35%) or living near landfill (30% vs. 81%). 

In Table 2, authors affirmed that most waste reclaimers do not wear dust mask at Landfill 1, and some were using garments or scarves to protect the respiratory tract from dust and chemicals. This is contradictory with the high rates of PPE use reported, and authors should explain it better.

Table 4: The adjustment variables must be described as table footnote. Variables with significant differences between groups such as  sex, education level, and chemical exposure, were not considered as confounding factors in multivariate regression models. It can be due to these variables were above the 5% level considered, but it can be better explained in the methods section. Regarding age, authors used 51+ as reference but wouldn't be more logical to use lower ages as reference considering that they have less cumulative exposure time? The same happens to "Perception that landfill affects chest", and "airborne dust". I would suggest to consider always the lesser exposed as reference group. There are missing associations in the adjusted models (e.g. age, education, Tb prevalence). Data presentation in this table can be improved (e.g. both headings and sub-headings in bold). I would suggest to exclude variables with less than 5 observations of this table (e.g. use of gas or other sources of fuel). 

In the text, authors stated that respiratory symptoms of site 1 workers had an adjusted OR = 2.77 (95% CI 0.96- 7.01), which is not significant, but different values were presented in the table (2.77; 1.03; 7.42). Please, check for consistence. The statement "current smokers and previous history of smoking showed a significant association with respiratory symptoms" but this is not supported by your data (CI including 1.0). 

Discussion: Authors affirmed that "socio-demographic characteristics such as living adjacent to the landfill sites, number of years a waste recycler, use of biomass as a source of fuel did not show a significant association with reporting respiratory symptoms". It is not clear whether they refer to crude or adjusted models. Also, if they tested the associations for living near landfill sites, it is not presented in table 4.  

Authors should critically discuss they findings (e.g. symptoms prevalence) comparing with other studies, and not only present them. What are the differences and similarities?

Two references are given to the statement: "Chokandre et al (2017) found a high prevalence of difficulty in breathing and a chronic cough among the waste-pickers of Mumbai in India [13, 14]."

Authors mixed the presentation of symptoms with exposure factors (lines 193-197 with 198-200), and it can confound the readers.

The statement: "The prevalence of muscle aches and sprains was 29%, compared to other studies that reported a high prevalence of musculoskeletal disorders in Brazilian waste recyclers [4]." (lines 203-205) seems incomplete.

Authors affirmed that: "the odds of reporting respiratory symptoms for those with a history of smoking was almost four times, in addition smoking was found to be marginally associated with increased prevalence of respiratory symptoms." (lines 213-215), but seems like there is a confusion between the findings of crude and adjusted models, and statistical significance (CI).     

Authors discussion about the main associations observed is presented lately in the discussion section, respectively, landfill site (lines 218-227) and chemical use (lines 228-234). My suggestion is that they should consider presenting them first. 

The term "water cyclers" used in line 221 was not previously presented or explained elsewhere in the text. 

The statement: "The lack of awareness to use proper protective clothing
might be attributed to the lack of education on occupational health of the waste recyclers and on the dearth of knowledge on hazard associated with waste picking." (lines 225-227) can be supported by a reference.

The statement: "The analysis showed that landfill workers exposed to airborne dust reported twice the odds of respiratory symptoms than those not exposed." (lines 235-236) is not supported by your findings because of the non-significant associations and large CI.

Regarding study limitations: "a quantitative phase of the study is
recommended to look at respiratory health. The spirometric lung function measurements and environmental exposure assessments could have been used to strengthen the reliability of the study results. Lastly, the smaller sample size is also a limitation in predicting the statistical effect of the independent variables." (lines 246-251) can be improved. Some insights are: a) as this study had a quantitative part, authors should recommend a complementary study (if possible with spirometry and exposure assessment) to strengthen their findings; and b) although the study's sample size can be improved, it was very close to the 365 previously calculated as sufficient, and can be reported as strength of the study. 

Author Response

Reviewer 3

Comment

Addressed (Yes/No)

comment is addressed/Justification

Page and line number

Introduction

Replace “developing countries” by low-income or low-and middle-income countries.

Yes

Replaced “in low and middle income countries”

Page 1 of 12, line 35.

Method and results

Authors alternate the data presentation e.g. n (%) and % (n). It must be standardized

Yes

Corrections done in the results section. data is presented as n (%).

Page 3 of 12, line 128-143.

Abstract

Specify which chemicals waste recyclers are exposed to and how they are exposed.

Yes

Indicated in the methods the type of chemical waste, recyclers are exposed to. “Exposure to chemical waste (cleaning detergents, paint, gases)”

Page 3 of 12, line 104-105.

Methods/ data analysis

Variables that are presented in the final model must be described in the methods.

Specify what the use of PPE is refereed to? Respiratory PPE, any PPE or at least one?

Yes

Yes

The following paragraph was added to the methods section

“The questionnaire was made up of the following data variables: socio-economic variables: collecting data on personal information pertaining to the participants, such as age, sex, education, number of years working as a waste recycler, living near a landfill site and source of fuel for cooking which included whether they used electricity, paraffin, gas, wood/coal and any other source. Behavioral characteristics included a history of cigarette smoking and number of years smoking. The waste recyclers were asked whether they thought working at a landfill site affects their chest. The question was phrased as “Do you think working at a landfill site affects your chest “Yes or No”. information was also collected on their medical history, including previously being diagnosed with TB and asthma and whether they experienced any muscle sprains and strains.

Specified in the methods “The use of PPE (mask, boots or gloves) was classified as (0) = “always”, (1) = “sometimes or never”

Page 2 of 12, line 88-96.

Page 3 of 12, line 102-103.

Results

P values of 0.000 must be replaced by p<0.001.

The variable “landfill affect chest” is not described and explained in the methods.

Yes

Yes

Replaced p values 0.000 with p < 0.001

Addressed in the methods section. “The waste recyclers were asked whether they thought working at a landfill site affects their chest. The question was phrased as “Do you think working at a landfill site affects your chest “Yes or No”

Line 145.

Page 3 of 12, line 94.

In table 2, authors affirmed that most waste recyclers do not wear dusk mask at landfill 1, and some wear seen using garments and scarves. This is contradictory with the high rates of PPE used. Authors should explain.

No

An observation was made during the walkthrough assessment were the waste recyclers were seen using garments and scarves as they protective clothing for the work. However when asked during the interview they specified using PPE during their work.

Table 4 :

The adjustment variables must be described as table footnote.

Variables with significant differences between groups such as sex, education level, and chemical exposure, were not considered as confounding factors in multivariate regression models. It can be due to these variables were above the 5% level considered, but it can be better explained in the methods section.

Yes

Yes

Included in the table footnote.

Addressed in the methods section. Model building was used to choose variables that were included in the final model. Variables that were statistically significant and those that are considered to be important based on existing literature were included in the final model. Those that were not significant in the Univariable analysis were excluded in the final model (sex, education level)

Line 174-175.

Line 118-120.

Table 4:

Data presentation in this table can be improved (e.g. both headings and sub-headings in bold).

Yes

Table formatted.

In the text, authors stated that respiratory symptoms of site 1 workers had an adjusted OR = 2.77 (95% CI 0.96- 7.01), which is not significant, but different values were presented in the table (2.77; 1.03; 7.42). Please, check for consistence

Yes

Correction made in the text

Page 4 of 12, line 178.

The statement "current smokers and previous history of smoking showed a significant association with respiratory symptoms" but this is not supported by your data (CI including 1.0).

Yes

…. corrected

“…while those who reported being current smokers and previous history of smoking showed a marginally significant association

Page 4 of 12, line 181

Discussion

Authors affirmed that "socio-demographic characteristics such as living adjacent to the landfill sites, number of years a waste recycler, use of biomass as a source of fuel did not show a significant association with reporting respiratory symptoms". It is not clear whether they refer to crude or adjusted models.

Also, if they tested the associations for living near landfill sites, it is not presented in table 4. 

Yes

Yes

Amended to “In the adjusted multivariable analysis socio-demographic characteristics (age, sex, educational level) and other predictors such as the number of years working as a waste recycler, source of fuel for cooking and use of PPE did not show a significant association with reporting respiratory symptoms..

Removed in the final analysis as it was not significant.

Page 4 of 12, line 192-195.

Authors should critically discuss they findings (e.g. symptoms prevalence) comparing with other studies, and not only present them. What are the differences and similarities?

Yes

Studies from other studies have been used to present similarities with our results and also differences that were observed.

“… Similarly, Chokandre et al (2017) found a high prevalence of difficulty in breathing and a chronic cough among the waste-pickers of Mumbai in India [15]. Oyelola et al (2011) also found coughing to be the major respiratory health problem for waste pickers working at landfill sites in Lagos metropolis [16]”.

Line 198-204.

Two references are given to the statement: "Chokandre et al (2017) found a high prevalence of difficulty in breathing and a chronic cough among the waste-pickers of Mumbai in India [13, 14]."

Yes

Reference has been corrected.

Page 4 of 12, line 201.

Authors mixed the presentation of symptoms with exposure factors (lines 193-197 with 198-200), and it can confound the readers.

Yes

 Statement in line 193-197 and 198- 200 have been removed.

Authors affirmed that: "the odds of reporting respiratory symptoms for those with a history of smoking was almost four times, in addition smoking was found to be marginally associated with increased prevalence of respiratory symptoms." (lines 213-215), but seems like there is a confusion between the findings of crude and adjusted models, and statistical significance (CI).    

No

The statement refers to the adjusted models. An odds ratio of 3.52 (95% CI: 0.92; 15.5, p=0.066). the p value does not indicate a significant association but rather marginally significant and was thus retained in the final model.

Table 4.

Authors discussion about the main associations observed is presented lately in the discussion section, respectively, landfill site (lines 218-227) and chemical use (lines 228-234). My suggestion is that they should consider presenting them first.

Yes

Main observations of the study moved to the beginning of the discussion.

Page 4 and 5 of 12.

The term "water cyclers" used in line 221 was not previously presented or explained elsewhere in the text.

Yes

The word has been fixed.

..”waste recyclers…”

Page 5 of 12, line 232.

The statement: "The lack of awareness to use proper protective clothing

might be attributed to the lack of education on occupational health of the waste recyclers and on the dearth of knowledge on hazard associated with waste picking." (lines 225-227) can be supported by a reference.

Yes

Reference added at the end of the statement.

Line 238.

The statement: "The analysis showed that landfill workers exposed to airborne dust reported twice the odds of respiratory symptoms than those not exposed." (lines 235-236) is not supported by your findings because of the non-significant associations and large CI.

Yes

…..

Despite our findings not being statistically significant, related studies found that the majority of waste pickers complained of bad smell and dust from the landfill which they indicated affects their health [23, 24].

Line 212-213.

Regarding study limitations: "a quantitative phase of the study is.. (lines 246-251) can be improved. Some insights are: a) as this study had a quantitative part, authors should recommend a complementary study (if possible with spirometry and exposure assessment) to strengthen their findings; and b) although the study's sample size can be improved, it was very close to the 365 previously calculated as sufficient, and can be reported as strength of the study.

Yes

Suggestion put into the discussion.

“Recall bias may also have affected the estimated prevalence of the outcome; thus authors recommend a complementary study with spirometry and exposure assessment to strengthen the findings in this study. Even though the study's sample size can be improved, it was very close to the 365 previously calculated as sufficient, and can be reported as a strength of the study.

Page 5 of 12 , line 239-246.

Reviewer 4 Report

Thank you for the good and very relevant work! I think you can further improve it if addressing the following items:

more details about waste stored at two studied landfills? Is there any information about type of the waste there? Is it municipal waste, or includes industrial waste, etc.... was the assessment of chemicals on landfills base don subjective answer of respondents only? was there any verification done on this issue? quite high level of smoking among respondents; I would expect more analytical data and also discussion on role of smoking as confounder or effect modifier in this study? Do people smoke while working at landfill? prevalence of respiratory symptoms is very different on two landfills; why is it so? some data in Results is duplicated in text and tables. It would be easier for reader if data is concentrated in Tables and there is less text in Results. You are welcome to add to discussion rather.

Author Response

Reviewer 4

Addressed (Yes/No)

comment is addressed/Justification

Page and line number

Addressed (Yes/No)

Provide more details about waste stored at two studied landfills? Is there any information about type of the waste there? Is it municipal waste, or includes industrial waste, etc

Yes

There are four Pikitup landfill sites in Johannesburg, Gauteng, South Africa where municipal waste is disposed.

Page 2 of 13, line 68.

was the assessment of chemicals on landfills based on subjective answer of respondents only? was there any verification done on this issue?

Yes

Assessment of chemicals on landfill site was based on participant’s response on whether they were exposed or not and whether or not they were in contact with chemical waste during waste sorting.

quite high level of smoking among respondents; I would expect more analytical data and also discussion on role of smoking as confounder or effect modifier in this study?

Yes

Included in discussion “Smoking increases the prevalence of respiratory symptoms [5, 17]. The prevalence of smoking is higher in low socioeconomic groups especially disadvantaged individuals such as the informal workers [18, 19]. In our analysis, the odds of reporting respiratory symptoms for those with a history of smoking was almost four times, in addition smoking was found to be marginally associated with increased prevalence of respiratory symptoms. Smoking is a well-known confounder of respiratory symptoms [5, 20].

Page 5 of 13 , line 222-228.

Do people smoke while working at landfill?

No

The comment was considered. Nonetheless, during the walkthrough we did not observe waste recyclers that were smoking at the landfill site

prevalence of respiratory symptoms is very different on two landfills; why is it so?

Yes

Indicated in the discussion. “There were differences in work practices and the condition of work between the two landfill sites observed during the risk assessment walkthrough”.

Page 5 of 13, line 231-232.

some data in Results is duplicated in text and tables. It would be easier for reader if data is concentrated in Tables and there is less text in Results.

Yes

Comment has been considered. Data is included in the text to direct the reader to the justification of the results that are indicated in the tables.

Round 2

Reviewer 1 Report

Thank for the corrections. I do not have any further comments. 

Author Response

done

Reviewer 2 Report

The authors have addressed most of my concerns.

Some of my concerns were either misunderstood or not taken into account.

Nevertheless, I think that the ms is now of suitable quality for the journal.

Author Response

done

Reviewer 3 Report

After a careful analysis of the new version and answers provided, I consider that the authors improved the manuscript based on the reviewer's comments, and now the paper is ready to be published. Best Regards.

Author Response

done